biomechanics/biomedical engineering

eye, cornea, OCT, orthokeratology, contact lenses, epithelium

**Authors for correspondence:**
Ziying Ran
e-mail: z.ran@liverpool.ac.uk
Jun Jiang
e-mail: jjhsj@hotmail.com

# A new approach for quantifying epithelial and stromal thickness changes after orthokeratology contact lens wear

Ziying Ran[1], Joshua Moore[1,2], Fan Jiang[3], Hongmei Guo[4], Ashkan Eliasy[1], Bernardo T. Lopes[1,5], FangJun Bao[3], Jun Jiang[3], Ahmed Abass[1,6] and Ahmed Elsheikh[1,7,8]

[1]School of Engineering, and [2]Department of Mathematical Sciences, School of Physical Sciences, University of Liverpool, Liverpool, UK
[3]Eye Hospital, Wenzhou Medical University, Wenzhou, People's Republic of China
[4]College of Biomedical Engineering, Taiyuan University of Technology, Taiyuan City, Shanxi Province, People's Republic of China
[5]Federal University of São Paulo, 1500 Vila Clementino, São Paulo 04021-001, Brazil
[6]Department of Production Engineering and Mechanical Design, Faculty of Engineering, Port Said University, Port Fuad, Egypt
[7]Beijing Advanced Innovation Centre for Biomedical Engineering, Beihang University, Beijing, People's Republic of China
[8]National Institute for Health Research (NIHR) Biomedical Research Centre at Moorfields Eye Hospital NHS Foundation Trust and UCL Institute of Ophthalmology, London, UK

ZR, 0000-0002-2562-5109; AA, 0000-0002-8622-4632

The aim of the study was to develop an automatic segmentation approach to optical coherence tomography (OCT) images and to investigate the changes in epithelial and stromal thickness profile and radius of curvature after the use of orthokeratology (Ortho-K) contact lenses. A total of 45 right eyes from 52 participants were monitored before, and after one month of, uninterrupted overnight Ortho-K lens wear. The tomography of their right eyes was obtained using optical OCT and rotating Scheimpflug imaging (OCULUS Pentacam). A custom-built MATLAB code for automatic segmentation of corneal OCT images was created and used to assess changes in epithelial thickness, stromal thickness, corneal and stromal profiles and radii of curvature before, and after one month of, uninterrupted overnight wear of Ortho-K lenses. In the central area (0–2 mm diameter), the epithelium thinned by $12.8 \pm 6.0$ µm (23.8% on average, $p < 0.01$) after one month of

Ortho-K lens wear. In the paracentral area (2–5 mm diameter), the epithelium thinned nasally and temporally (by $2.4 \pm 5.9$ µm, 4.5% on average, $p = 0.031$). The stroma thickness increased in the central area (by $4.8 \pm 16.1$ µm, $p = 0.005$). The radius of curvature of the central corneal anterior surface increased by $0.24 \pm 0.26$ mm (3.1%, $p < 0.01$) along the horizontal meridian and by $0.34 \pm 0.18$ mm (4.2%, $p < 0.01$) along the vertical meridian. There were no significant changes in the anterior and posterior stromal radius of curvature. This study introduced a new method to automatically detect the anterior corneal surface, the epithelial posterior surface and the posterior corneal surface in OCT scans. Overnight wear of Ortho-K lenses caused thinning of the central corneal epithelium. The anterior corneal surface became flattered while the anterior and posterior surfaces of the stroma did not undergo significant changes. The results are consistent with the changes reported in previous studies. The reduction in myopic refractive error caused by Ortho-K lens wear was mainly due to changes in corneal epithelium thickness profile.

## 1. Introduction

The prevalence of myopia has increased rapidly in recent years. In 2010, there were 1950 million people with myopia globally, and the number is expected to increase to 4758 million, or 49.8% of the population, by 2050 [1]. Orthokeratology (Ortho-K) is a clinical technique that temporarily reshapes the cornea to eliminate or reduce the refractive error [2]. Ortho-K lenses are commonly worn during sleep and removed immediately after waking up, allowing clear unaided vision in daytime. The lenses are made of rigid gas permeable (RGP) material and seek to reduce myopia by corneal flattening [3]. The base curve of the lens is flatter than the central corneal radius of curvature and the secondary curve (in the paracentral region) is steeper than the base curve [4]. Earlier studies showed that Ortho-K contact lenses effectively reduce myopia, providing reliable refractive correction up to 4.5 D [5,6]. High myopia has been partially reduced by Ortho-K lenses in some studies [7,8]. Yet, despite interest in Ortho-K lenses, the mechanism by which they interact with the cornea is still not fully understood. In several studies, epithelial thickness changes were recognized as the main cause of corneal reshaping [9–16].

Various methods have been used to measure the geometrical changes that take place in the cornea as a result of wearing Ortho-K lenses, and in particular the thickness changes in the stroma and epithelium, and the radius of curvature changes in the cornea's anterior and posterior surfaces. The rotating Scheimpflug camera provides reliable three dimensions, non-contact imaging of corneal tomography [17], and has good reproducibility and repeatability in measuring radii of curvature and thickness in both normal and keratoconic eyes [18]. Using Scheimpflug imaging (the Pentacam HR system), Gonzalez-Mesa et al. [19] reported significant reductions in anterior chamber depth and flattening in the posterior corneal surface in 34 subjects after one year of wearing Ortho-K lenses. Anterior segment optical coherence tomography (OCT) was also used to measure stromal and epithelial thickness profiles [20–22] with higher scanning speed and better resolution relative to Scheimpflug systems [21,22]. With OCT, Neito-Bona et al. [23] measured the epithelial thickness in 27 subjects before and one month after wearing Ortho-K lenses, and showed significant epithelial thickness reductions in the central cornea up to 2 mm diameter. Similar observations, and also epithelial thickening in the peripheral region, were made based on analysis of OCT corneal scans [11,15,16,24].

Different methods to detect epithelial and stromal surfaces in OCT images were reported in the literature. Elsawy et al. [25] used Randomized Hough Transform to develop an OCT image segmentation algorithm, including estimations of corneal thickness. The resulting automatic segmentation of 15 corneal OCT images was compared with manual segmentation by two trained operators, and the mean difference between the automated and manual methods was $5.66 \pm 6.38$ µm. Eichel et al. [26] detected the anterior and posterior surfaces of the cornea automatically using the difference in refractive index between the cornea and both the air and aqueous, and used a low-dimensional function to describe the cornea's three inner layers. Elsawy et al. [25] also detected the cornea's outer surfaces and transformed them into a flat form to locate the inner layers. Compared with manual segmentation, this method has better repeatability and similar precision.

While these methods were largely successful in tracing the anterior and posterior surfaces, difficulties remained in detecting the inner surfaces, and some methods involved manual segmentation steps. These points were the drivers to develop a new automatic segmentation method in this study that can detect with high repeatability both outer and inner corneal surfaces.

# 2. Material and methods

## 2.1. Study participants

This prospective study included 52 participants who were prescribed Ortho-K lenses for myopia correction in the Eye Hospital of Wenzhou Medical University between December 2016 and August 2019. The inclusion criteria included age between 8 and 18 years, best-corrected visual acuity (BCVA) greater than 20/25, spherical equivalent between −1.00 D and −6.00 D and intraocular pressure between 11 and 21 mmHg. Participants with history of Ortho-K lens wear, contact lens contraindications, related eye systemic diseases, uncorrected visual acuity below 20/25, or unacceptable lens decentration (over 1 mm) after regular wear were excluded. These inclusion and exclusion criteria, along with details of the clinical care adopted, follow those presented in our earlier clinical study [27]. The study was approved by the Ethics Committee of the eye hospital and complied with the Declaration of Helsinki. Examinations were only conducted after the subjects and their guardians had fully understood and signed informed consent.

## 2.2. Lens fitting

Participants were fitted with Ortho-K lenses produced by Euclid Systems Corp., USA, Paragon CRT, USA and Dreamlite, Germany. The Euclid lenses, manufactured with Boston Equalens II material, had Dk of $85 \times 10^{-11}$ $(cm^2\,s^{-1})$ $(mLO_2/ml \times mmHg)$, while the Paragon CRT lenses were made of Paragon HDS 100 Paflufocon D material with Dk of $100 \times 10^{-11}$ $(cm^2\,s^{-1})$ $(mLO_2\,ml^{-1} \times mmHg)$. Finally, the Dreamlite lenses were manufactured with the Boston XO hexafocon-A material that had Dk of $100 \times 10^{-11}$ $(cm^2\,s^{-1})$ $(mLO_2\,ml^{-1} \times mmHg)$. The lenses were fitted to both eyes by an experienced clinician (JJ) according to the manufacturer's guidelines, and participants were requested to wear the lenses for eight hours each night except in cases of illness or abnormal ocular symptoms [27,28].

## 2.3. OCT and Pentacam

A custom-built ultra-high-resolution, spectral-domain OCT (UHR-SD-OCT) with 3 μm axial resolution, mounted on a standard slit-lamp, was used to measure corneal tomography. The OCT acquired images with 24 k A-lines per second, and each B-scan included 2048 × 1365 pixels, corresponding to a scan width of 8.93 mm in the air and a scan depth of 2.00 mm. Corneal profiles were then created from scans of the vertical and horizontal meridians. A Pentacam HR (Oculus, Wetzlar, Germany) with a Scheimpflug camera was used to scan the anterior segment of all eyes included in the study.

All OCT and Pentacam HR measurements were performed by an experienced operator (FJ) at baseline and after one month of Ortho-K lens wear. The measurements were taken between 9.00 and 11.00 to minimize the diurnal variation. The measurements were repeated until an examination quality of 'OK' was obtained [27,28].

After reviewing the medical records of the right eyes of the 52 study participants, seven were excluded because of the blurriness of OCT images, which resulted in failure to locate the epithelial surface. Therefore, a total of 45 eyes were included in the analysis.

## 2.4. Data processing

### 2.4.1. Apex detection

Each B-scan OCT image included 2048 × 1365 pixels, where the brightness of each pixel varied from 0 to 256. The cornea's apex was detected first for being the brightest region in both horizontal and vertical scans (figure 1). The centre of this region was set as the origin point of a Cartesian coordinate system with the patient's eye in the negative $Z$-direction. The nasal-temporal direction was assigned to the $X$-axis in horizontal scans, while the inferior–superior direction was assigned to the $Y$-axis in vertical scans.

### 2.4.2. Exclusion of data that is likely to be outside cornea

A binary weighting function based on the bi-conic model (equation (2.1)) was introduced to eliminate data above the upper boundary curve. The model was created by a binary weighting function and

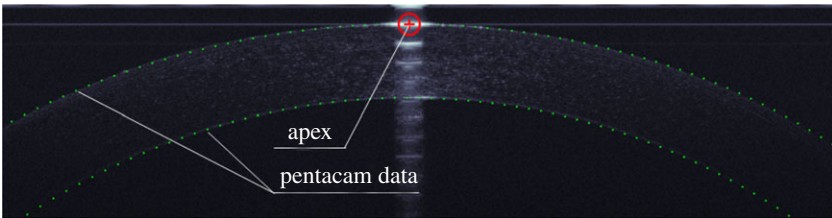

**Figure 1.** Apex detection (red cross and circle) and the Pentacam's elevation data (green dots) for the same corneal outer surfaces.

used to extend a circular equation by adding an asphericity coefficient, $Q_u$, to control the radius of curvature of the front boundary. A tolerance $T_u$ (with a value of 0.01 mm, which was found adequate in our preliminary study's trials) was added to the equation to shift the weighting function boundary slightly above the corneal anterior surface.

$$Z_u = T_u + \frac{\sqrt{R_u^2 - X_u^2(Q_u + 1)} - R_f}{Q_u + 1}. \tag{2.1}$$

The asphericity factor $Q_u$ is synonymous with the front boundary's radius of curvature, with positive values leading to increasing steepness away from the centre point, and negative values inducing a flattening effect. In this boundary, $Q_u$ was set to −0.2 [29,30] based on initial trials of the method. The parameter $R_u$ is the radius of the upper boundary, set to 10 mm to be considerably bigger than the cornea's range of anterior surface radius [29,30] to avoid any possibility of interference with that surface.

Similarly, a lower boundary was constructed with the similar conic model presented in equation (2.2). The tolerance $T_l$, set to −0.8 mm—substantially bigger in magnitude than normal values of corneal thickness [31]—was used to avoid conflict with the corneal posterior surface. Meanwhile, the lower boundary curve radius, $R_l$, was set to 5 mm with an asphericity coefficient $Q_l$ set to −0.1. Both values were found to be adequate in the study's initial trials. This choice of parameters avoided intersection between the boundary surfaces and the cornea's cross-section, while removing most of the unwanted portions of the image.

$$Z_l = T_l + \frac{\sqrt{R_l^2 - X_l^2(Q_l + 1)} - R_l}{Q_l + 1}. \tag{2.2}$$

### 2.4.3. Auto-detection of corneal surfaces

To reduce processing time, 129 evenly spaced Z-lines, parallel to the Z-axis, were considered including one aligned with the origin point of the Cartesian system. Surface detection was performed on only these lines.

Topography measurements of the corneal anterior surface along the principal vertical and horizontal meridians taken for the same corneas using the Pentacam (Oculus, Germany) (figure 1) were fitted to a 5th order polynomial. This polynomial order was sufficient to keep the root mean square of error of fit with the topography data below 2.5 µm, while avoiding overfitting. The polynomial was used to derive two profiles of the anterior surface along the vertical and horizontal meridians, and a search was then conducted to locate the point with the highest brightness along each of the 129 Z-lines that was within ±10 µm above or below the relevant profile. The resulting 129 points were fitted to a 5th order polynomial, and points that were away from the polynomial by 7 µm were considered outliers and were excluded before fitting the remaining points to another 5th order polynomial. This exercise was repeated until no outliers were identified, and this process was then repeated for the posterior surface.

### 2.4.4. Epithelium posterior surface detection

At least five points on the epithelium posterior surface were manually selected by the operator, including a point at each end of the visible width of the epithelium surface, and three points spaced roughly equally in between. These five points were fitted to a lower (4th) order polynomial to form an epithelium base curve, while avoiding overfitting. Points along the same 129 Z-lines (but only those lines between the endpoints marked above), that had the highest brightness and were within 10 µm from the base curve, were identified and assumed to be on the epithelial posterior surface. These points were fitted to a 5th order polynomial, and a process was followed to exclude points that were farther from the

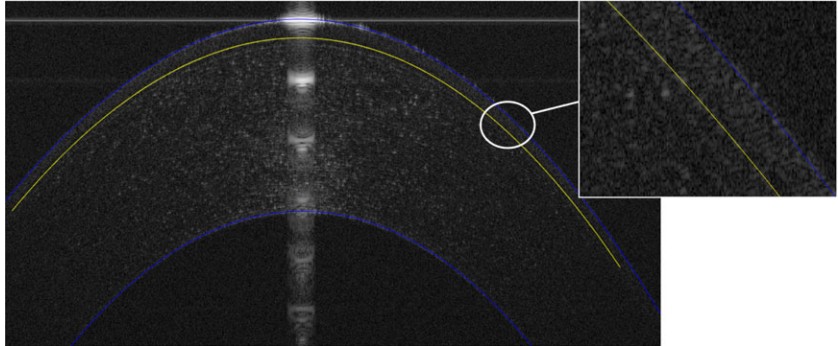

**Figure 2.** Automated segmentation result of corneal anterior, posterior and epithelial interfaces.

**Table 1.** Average epithelial, stromal and corneal thickness in central area (0–2 mm diameter).

| mean ± s.d. (µm) | before Ortho-K lens wear | after Ortho-K lens wear | p-value |
|---|---|---|---|
| epithelium | 53.6 ± 4.2 | 40.8 ± 5.3 | <0.01[a] |
| stroma | 515.7 ± 28.1 | 520.5 ± 27.6 | <0.01[a] |
| cornea | 569.3 ± 28.8 | 561.3 ± 28.8 | <0.01[a] |

[a]For Paired t-test $p < 0.01$.

**Table 2.** Average epithelial, stromal and corneal thickness in paracentral area (2–5 mm diameter).

| mean ± s.d. (µm) | before Ortho-K lens wear | | after Ortho-K lens wear | | p-value | |
|---|---|---|---|---|---|---|
| | horizontal meridian | vertical meridian | horizontal meridian | vertical meridian | horizontal meridian | vertical meridian |
| epithelium | 52.6 ± 4.3 | 51.6 ± 6.0 | 50.2 ± 4.9 | 53.9 ± 6.1 | 0.031 | 0.086 |
| stroma | 564.4 ± 29.9 | 565.5 ± 33.2 | 566.8 ± 28.5 | 569.1 ± 30.9 | 0.418 | 0.272 |
| cornea | 618.6 ± 32.0 | 625.9 ± 35.4 | 617.9 ± 29.8 | 624.8 ± 27.0 | 0.848 | 0.767 |

fitting curve by more than 7 µm. Fitting of the remaining points to a new polynomial and excluding the far points were repeated until no points could be excluded. The remaining points were then saved and used to represent the epithelial posterior surface.

### 2.4.5. Results confirmation

The lines representing the corneal anterior surface, corneal posterior surface and epithelium posterior surface were displayed on the OCT image to enable a visual check of all the fitting processes before accepting the results or repeating parts of the analysis which produced unsatisfactory results.

The epithelial thickness profiles in the horizontal and vertical scans were then adjusted proportionally to give the same thickness at the two central points ($X = 0$ mm), and the same process was followed for the stromal thickness profiles.

Once these steps were completed, the thickness profiles of both the stroma and epithelium were used to determine the radius of curvature profile along the three outer and inner surfaces in each scan, figure 2.

### 2.5. Data analysis

Data were recorded and analysed in MATLAB (v. 2020a for windows). All data were tested with the Kolmogorov–Smirnov normality test. Quantitative variables were expressed as mean ± standard deviation. Comparisons between pre-Ortho-K lens wear and post-Ortho-K lens wear were performed using the Paired t-test. Statistical significance was defined as $p < 0.05$.

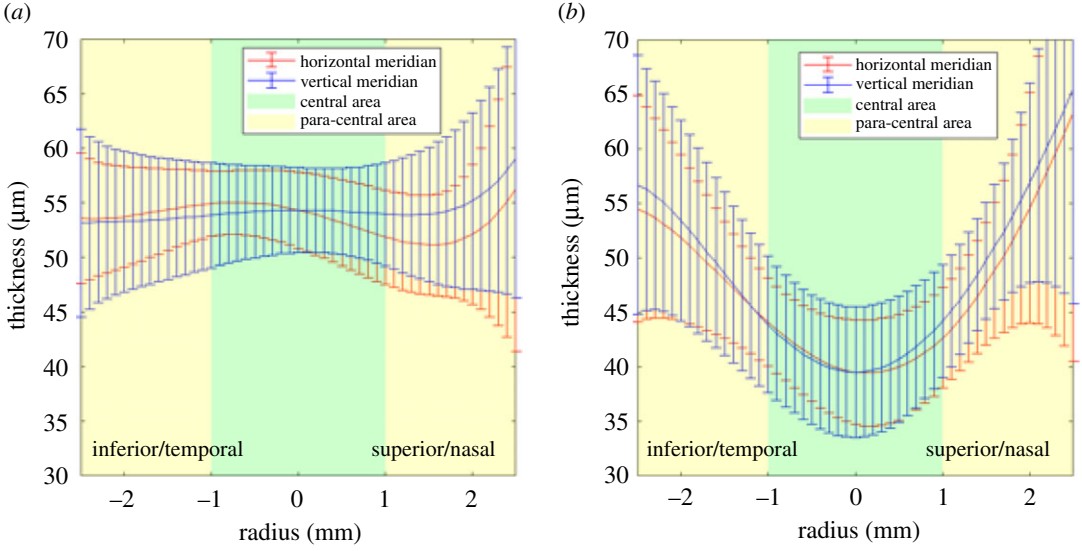

**Figure 3.** Mean epithelial thickness (*a*) before and (*b*) after Ortho-K lens wear.

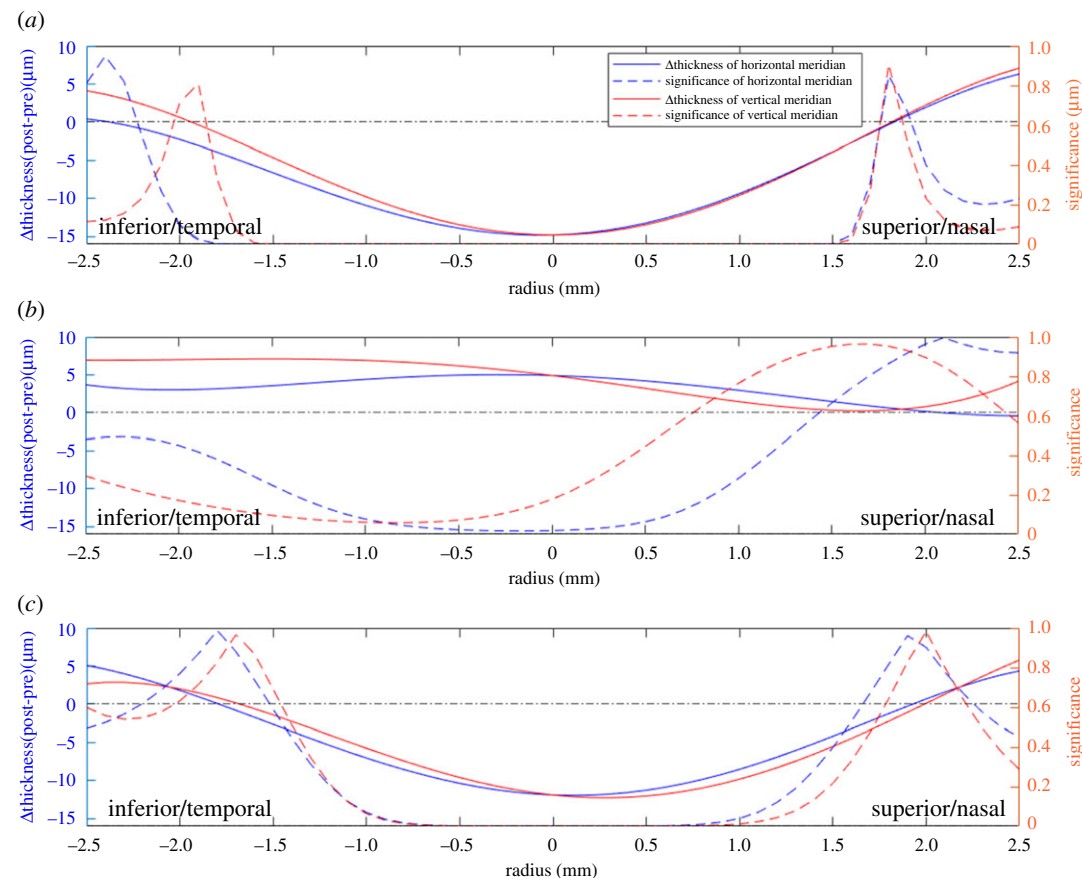

**Figure 4.** Values and significance of changes in thickness of (*a*) the epithelium, (*b*) the stroma and (*c*) the whole cornea.

## 3. Results

This study focused on the corneal area within 5 mm diameter because of the limitation in visible width of the epithelial surface on OCT images.

The thickness changes in the epithelium, stroma and the whole cornea were evaluated within the central area (0–2 mm diameter) and the paracentral area (2–5 mm diameter) in tables 1 and 2, figures 3 and 4. Overall, the epithelial thickness decreased by $12.8 \pm 6.0 \, \mu m$ (23.8% on average, $p <$ 0.01) in the central area, which was larger than the thickness changes in the paracentral area

**Table 3.** Central radius of curvature in OCT images, before and after orthokeratology lens wear.

| mean ± s.d. (mm) | before Ortho-K lens wear | | after Ortho-K lens wear | | p-value | |
|---|---|---|---|---|---|---|
| | horizontal meridian | vertical meridian | horizontal meridian | vertical meridian | horizontal meridian | vertical meridian |
| corneal anterior surface | 7.86 ± 0.27 | 7.60 ± 0.27 | 8.10 ± 0.21 | 7.94 ± 0.28 | <0.01[a] | <0.01[a] |
| epithelium posterior surface | 7.95 ± 0.31 | 7.58 ± 0.28 | 7.90 ± 0.29 | 7.57 ± 0.35 | 0.211 | 0.718 |
| stromal posterior surface | 6.50 ± 0.19 | 6.16 ± 0.24 | 6.51 ± 0.24 | 6.11 ± 0.24 | 0.780 | 0.139 |

[a]For Paired t-test $p < 0.01$.

(decreased $2.4 \pm 5.9$ µm, 4.5% on average, $p = 0.031$ along the horizontal meridian and increased $2.2 \pm 7.1$ µm, 4.2% on average, $p = 0.086$ along the vertical meridian). In the central area, the epithelium experienced thinning of up to 14.8 µm at the centre in the vertical scans ($p < 0.01$, figure 4), and up to 15.0 µm at −0.2 mm away from the centre in the horizontal scans ($p < 0.01$, figure 4). The significance of the thickness changes was close to 0 in the area from $X = -1.5$ mm to $X = 1.5$ mm in both meridians, then the changes became insignificant from a radius of 1.7 mm onwards.

For the stroma, the thickness increased moderately in both the central and paracentral areas. The thickness increase was $4.8 \pm 16.1$ µm ($p < 0.01$) in the central area, while in the paracentral area, the increase was $3.6 \pm 20.0$ µm ($p = 0.272$) along the vertical meridian, and $2.4 \pm 17.2$ µm ($p = 0.418$) along the horizontal meridian. In these results, the mean changes in stromal thickness were smaller than corresponding changes in epithelial thickness ($p < 0.05$), while the standard deviation values in the epithelium were significantly smaller than in the stroma ($12.8 \pm 6.0$ µm versus $4.8 \pm 16.1$ µm, $p < 0.01$), indicating much more consistent changes in epithelial thickness. The stromal thickness changes were significant at $X = -0.5$ mm to $X = 0$ mm in the horizontal meridian (all $p < 0.05$), while elsewhere the changes were not significant.

The overall thickness changes in the whole cornea indicated similar trends in the horizontal and vertical scans, including a decrease in the central area and an increase near the edges of the paracentral area. The cornea thinned within the central area by up to 11.9 µm horizontally ($p < 0.01$, figure 4) and up to 12.2 µm vertically ($p < 0.01$, figure 4). These changes were dominated by the thickness changes observed in the epithelium, which accounted for more than 60% of the total corneal thickness changes.

The central radius of curvature of the cornea's three surfaces before and after wearing Ortho-K lenses are compared in table 3. The anterior surface experienced a significant increase in central radius (indicating a flattened anterior surface) after lens wear in both horizontal and vertical scans—the radius increase was $0.24 \pm 0.26$ mm (3.1%, $p < 0.01$) along the horizontal meridian and $0.34 \pm 0.18$ mm (4.2%, $p < 0.01$) along the vertical meridian. By contrast, the epithelium posterior surface and the stromal posterior surface experienced slight and non-significant changes in the central radius of curvature after lens wear.

# 4. Discussion

In this study, we used ocular anterior segment imaging by the OCT to determine the thickness of the epithelium, the stroma and the total cornea in Ortho-K lens wearers before, and after one month of, overnight lens wear. The measurements were taken across the centre of the cornea along both the horizontal and vertical meridians. The OCT image analysis indicated a significant reduction in central epithelial thickness by $14.8 \pm 5.9$ µm (or 26% of initial thickness) attributable to Ortho-K treatment ($p < 0.01$). These results are consistent with earlier reports in which the central epithelial thickness decreased over 30 days of lens wear by $15.8 \pm 3.3$ µm in 18 eyes [12], and by up to 18 µm in one eye [32]. Other studies reported epithelium thickness reduction after one week of lens wear of $10.6 \pm$

4.2 µm [10], and 6.1 ± 1.6 µm [15]. Others adopted a one-day follow up and reported epithelial thickness reductions of 4.6 ± 2.7 µm [10], and 8.7 ± 4.8 µm [12].

By contrast, the epithelial thickness increased in the paracentral area following Ortho-K lens wear, but the increases were significantly lower than the decreases observed in the central area. The thickness increases in the paracentral area could be due to the deformation of epithelial cells caused by the negative pressure exerted by the Ortho-K lens in the reverse curve zone [23,33], and remodelling of the epithelium due to interference by the contact pressure with the normal centripetal migration of epithelial cells from the limbus could be another reason. The paracentral thickness increases observed in our study are compatible with the mean values reported by Zhang *et al.* [15], in which epithelial thickness increases were noted in both the paracentral and peripheral areas, albeit the change did not reach statistical significance. However, Zhang also mentioned that the lack of significance of changes in the paracentral area may have been due to variations in the size and location of reshaping areas among individuals.

The present study also noted different epithelial thickness changes along the horizontal and vertical meridians. Along the horizontal meridian, the point with the largest thinning was 0.2 mm temporally away from the centre, while it was at the centre in vertical scans. In the paracentral area, the epithelium thickened less on the temporal side than on the nasal side, while the thickness changed evenly in vertical meridians. As Lian *et al.* [11] has explained, these differences may be due to variations in the eyelid pressure acting in different eye regions. They could also be due to different lens-on-eye decentrations among individuals [15].

Stromal thickness experienced small, but significant, increases in the central area with a large scatter in values. These findings are compatible with Reinstein's results [32], in which the stromal thickness was 5 µm larger, on average, after Ortho-K within the central 3 mm diameter area. In the paracentral area, the stromal thickness experienced only small and non-significant variations after lens wear. These results are similar, to some extent, to a report by Nieto-Bona *et al.* [23] of central stromal thinning and thickening in paracentral area after one month of lens wear although these did not reach statistical significance, and another by Alharbi & Swarbrick [12] showing stromal thickness increases in the paracentral area over three months of lens wear. While oedema by overnight hypoxia is a likely reason for central thickening, paracentral thickness changes could be due to the combined effect of oedema and the negative pressure in the reverse curve zone, as noted by Kim *et al.* [16]. Overall, the mechanism behind stromal thickness change is still unclear, and further work is necessary to understand the effect of Ortho-K lens wear. However, the relatively small changes in stromal thickness mean that the trends observed in overall corneal thickness were dominated by the epithelial thickness changes. This finding agrees with those reported by Zhang *et al.* [15], and Lian *et al.* [11].

Our study also presented evidence that the corneal anterior surface experienced significant flattening in the central region after Ortho-K lens wear. Aligned with the results of this study, Masseedupally *et al.* [34] found that, after 14 days of Ortho-K lens wear, the cornea became significantly flattened, and this flattening was greater on the temporal side than the nasal side. Another study by Queiros *et al.* [35] reported central anterior surface flattening of 3.16 ± 1.78 D after 12 months of lens wear, and similar results were reported in Chen *et al.*'s study [10]. However, limited research was conducted to study posterior corneal radii of curvature. Yoon *et al.* [36] reported no significant changes in either posterior corneal apical radius or asphericity after 14 days of lens wear. In our study, the central epithelial posterior surface and the stromal posterior surface showed no significant radii of curvature changes after Ortho-K lens wear. These results demonstrated that the Ortho-K lens mainly affected the epithelium and the corneal anterior surface, a result that is in agreement with previously published data [35,37].

In this study, we established a method to automatically detect the three external and internal corneal surfaces in OCT scans. The surface detection results for 45 patients (180 OCT images) indicated the suitability of this method in segmenting corneal images. The method builds on the success of previous work [25,38,39], and provides advantages in automation and validation in a relatively large patient population. To further enhance the new method, denoising of original OCT images and auto-detection of other corneal interfaces should be considered in future work.

This study has a number of limitations. First, the peripheral region of the cornea could not be examined as most OCT images lacked clarity in this region. Second, the study defined stromal thickness as total corneal thickness minus epithelium thickness. Thus, stromal thickness in our study included the Bowman membrane, Descemet membrane and endothelium, and these layers could not be separated in the image analysis. Finally, variations in the follow up time after commencing lens wear, due to booking restrictions and participants' personal issues (34 ± 6 days, range: 27–43), may have caused some variability in changes in thickness and radius of curvature.

In conclusion, this study introduced a new custom-built method to automatically detect the anterior corneal surface, the epithelial posterior surface and the posterior corneal surface in OCT scans. Overnight wear of Ortho-K lenses caused thinning of the central corneal epithelium. The anterior corneal surface became flattened while the anterior and posterior surfaces of the stroma did not undergo significant changes. The reduction in myopic refractive error caused by Ortho-K lens wear was mainly due to changes in corneal epithelium thickness profile.

Ethics. This prospective study was approved by the Ethics Committee of the Eye Hospital of Wenzhou Medical University. The ethical approval number is KYK [2015] 29. The inclusion criteria were as follows: 8–18 years of age, spherical equivalent from –1.00 D to –6.00 D, best-corrected visual acuity greater than 20/25, intraocular pressures were within normal limits (11–21 mmHg), no history of Ortho-K lens wearing, no contact lens contraindications or related eye and systemic disease. Patients who had uncorrected visual acuity less than 20/25 or unacceptable lens decentration after regular wear for one month were excluded from the study. All procedures complied with the Declaration of Helsinki. All the examinations were conducted after the participants and their guardians fully understood and signed informed consent.

Data accessibility. This article has no additional information.

Authors' contributions. Z.R.: Formal analysis, software, visualization, writing—original draft, writing—review and editing; J.M.: formal analysis, writing—review and editing; F.J.: resources, writing—original draft, writing—review and editing; H.G.: writing—original draft; A.E.: data curation, validation; B.T.L.: data curation, validation; F.J.B.: resources; J.J.: resources; A.A.: conceptualization, methodology, project administration, software, supervision, writing—original draft, writing—review and editing; A.E.: conceptualization, methodology, project administration, supervision, writing—review and editing. All authors gave final approval for publication and agreed to be held accountable for the work performed therein.

Competing interests. We declare we have no competing interests.

Funding. This research received no specific grant from any funding agency in the public, commercial, or not-for-profit sectors.

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
