## [Peer Review File · Royal Society Open Science]

Review History

RSOS-211108.R0 (Original submission)

Review form: Reviewer 1

Is the manuscript scientifically sound in its present form?

Yes

Are the interpretations and conclusions justified by the results?

Yes

Is the language acceptable?

Yes

Do you have any ethical concerns with this paper?

No

Have you any concerns about statistical analyses in this paper?

No

Recommendation?

Accept with minor revision (please list in comments)

Comments to the Author(s)

This is a solid piece of work that clarifies the mechanisms underlying the temporary changes induced by orthokeratology. My comments are minor.

Lines 138-140: numbers don't add up

Line 160: "curvature" is the inverse of radius. I suggest a different word be used.

Review form: Reviewer 2

Is the manuscript scientifically sound in its present form?

Yes

Are the interpretations and conclusions justified by the results?

No

Is the language acceptable?

Yes

Do you have any ethical concerns with this paper?

No

Have you any concerns about statistical analyses in this paper?

Yes

Recommendation?

Major revision is needed (please make suggestions in comments)

Comments to the Author(s)

In this study, 52 patients wore orthokeratology lenses in an overnight wear modality for up to one month. A custom-built MATLAB code was used for semi-automatic segmentation of corneal OCT images in the two principal meridians out to a 5-mm chord, and changes in epithelial and stromal thickness after lens wear were measured. Central epithelial thinning was found to explain most of the refractive effect of orthokeratology, and some paracentral thickening was also noted. No significant changes in anterior or posterior stromal radius of curvature were found.

Overall, this is an interesting study, but presents very similar results as have been published many times before over the last 20 years in terms of changes in corneal epithelial and stromal thickness, and radii of curvature of the various corneal layers resulting from orthokeratology. In other words, the findings are not new, and merely confirm the results of many previous studies. The contribution and novelty of the paper lie in the new methodology used to arrive at these results. In my view, the paper needs to be revised in terms of the Title (to emphasize the novelty of methodology) and in general presentation to emphasize the new measurement approach.

A recommendation for a new title might be "A New Approach for Quantifying Corneal Thickness Changes after Orthokeratology Contact Lens Wear". The focus of the abstract will need

to change significantly to highlight the new approach and to recognise that the findings are very similar to those reported in previous papers.

Throughout the paper, the authors refer to and discuss changes that do not reach statistical significance as if they are “real” changes. This over-interpretation of effects must be avoided, as failure to reach statistical significance means that any apparent changes may simply be due to chance. The manuscript should be carefully revised throughout to avoid this error of interpretation.

I have also made some minor comments below.

Page 1, line 14: This is a prospective study.

Abstract, lines 43-44: Were the changes in paracentral epithelial thickness statistically significant? If so, please insert p-values. If not, this needs to be noted. I also find it difficult to reconcile the data presented here with the data in Table 2 for both epithelial and stromal thickness changes.

Line 69: Two of these references (Charm et al and Kakita et al) demonstrate only partial correction of myopia in patients with higher degrees of myopia, and the other reference simply refers to Charm and Kakita – indeed the sentence written here “Subsequent studies... up to -10.0 D” is directly copied from the Bullimore reference! This sentence needs to be reconsidered.

Line 69: “Yes”?? The following sentence needs to be rewritten. It is grammatically mangled, and fails to take into account the many papers that have attempted to explain the biomechanical effects of orthokeratology.

Line 108: Please decide whether the participants were patients or subjects (or participants) and be consistent with terminology.

Line 114: Define “unacceptable lens decentration”.

Line 121: What lens material was used? It is likely to differ between the lens designs, and may not have a common Dk of 90.

There are many details about clinical care that are not included in the Methods, such as frequency of aftercares, lens solutions used, any lens parameter changes, any adverse effects etc. These details would be expected in a prospective clinical study such as this. This also further emphasizes that the main purpose of the paper was to apply the new methodology rather than to investigate the clinical impact of orthokeratology. Is it possible to refer to the clinical methodology previously published by some of the co-authors?

Lines 127 and following: It is not specifically stated if OCT images were obtained before and after one month of orthokeratology lens wear, and whether they were obtained between 9 AM and 11 AM. This is specifically stated for Pentacam data.

Lines 138-140: 52-7 does not equal 44 eyes – what happened to the other one?

Line 191: What was the visible width of the epithelial surface? I assume that this is what limited your analysis to a 5-mm chord?

Line 196: Is 10um correct? Or should line 190 explain that the “5 points” selected were on the posterior epithelial surface?

In Figure 3, please define nasal/temporal and superior/inferior directions.

Line 228: Is the negative value indicating the nasal or temporal meridian?

Line 242: The standard deviation values in the epithelium were significantly smaller than in the stroma. This would indicate more consistent epithelial thickness changes.

Line 281: Better to say “and after one month of overnight lens wear”.

Line 288: Spelling error: “lens”

Line 286-288: The current study was over one month of lens wear. This is a very different period from 1 night or even 1 week, as reported in the other studies referenced here (except for ref 30, which was after 30 days of lens wear). The effects over more comparable periods of lens wear would be more appropriate to cite here.

Line 294: Redistribution of epithelial cells out from the corneal centre to the paracentral area, against the normal centripetal migration of epithelial cells from the limbus, seems an unlikely explanation.

Line 309 and following: It is unclear how soon after eye opening/lens removal the stromal thickness changes were measured. Given the relatively low Dk/t of orthokeratology lenses, is it possible that this increase in stromal thickness may be attributable to overnight hypoxia?

Line 315: The Alharbi paper did not find stromal thickness increases across the cornea, only in the paracentral area.

Line 324 and following: See my earlier comment about comparing results from the current study (one month duration) with similar duration studies.

Line 330-331: Refer to study by Yoon et al (OVS 2013), which examined posterior corneal curvature changes in orthokeratology.

Line 337: Were there 44 or 45 participants?

Line 348: There has been no previous mention about variations in follow-up time. This needs to be reported somewhere – if the range was large, then this would obviously introduce some variability.

References: The formatting of the references is very poor and haphazard, and this needs to be tidied up by the authors. In many cases essential reference information including authors names, journal name, and journal details are missing or incorrect. Refer to the journal’s required style for reference formatting.

Decision letter (RSOS-211108.R0)

Dear Ms Ran,

The Editors assigned to your paper RSOS-211108 "Epithelial and Stromal Thickness Changes after Orthokeratology Contact Lens Wear" have now received comments from reviewers and would like you to revise the paper in accordance with the reviewer comments and any comments from the Editors. Please note this decision does not guarantee eventual acceptance.

Please submit your revised manuscript and required files (see below) no later than 21 days from today's (ie 04-Oct-2021) date. Note: the ScholarOne system will 'lock' if submission of the revision is attempted 21 or more days after the deadline. If you do not think you will be able to meet this deadline please contact the editorial office immediately.

on behalf of Dr Adil Al-Mayah (Associate Editor) and R. Kerry Rowe (Subject Editor)
openscience@royalsociety.org

Associate Editor Comments to Author (Dr Adil Al-Mayah):

The paper presents an interesting study with great potential contribution to the field. The novelty of the method to obtain the reported results is well recognized. However, as raised by one reviewer, some concerns regarding the novelty and validity of the results need to be address.

Reviewer comments to Author:

Reviewer: 1

Comments to the Author(s)

This is a solid piece of work that clarifies the mechanisms underlying the temporary changes induced by orthokeratology. My comments are minor.

Lines 138-140: numbers don't add up

Line 160: "curvature" is the inverse of radius. I suggest a different word be used.

Reviewer: 2

Comments to the Author(s)

In this study, 52 patients wore orthokeratology lenses in an overnight wear modality for up to one month. A custom-built MATLAB code was used for semi-automatic segmentation of corneal OCT images in the two principal meridians out to a 5-mm chord, and changes in epithelial and stromal thickness after lens wear were measured. Central epithelial thinning was found to explain most of the refractive effect of orthokeratology, and some paracentral thickening was also noted. No significant changes in anterior or posterior stromal radius of curvature were found.

Overall, this is an interesting study, but presents very similar results as have been published many times before over the last 20 years in terms of changes in corneal epithelial and stromal thickness, and radii of curvature of the various corneal layers resulting from orthokeratology. In other words, the findings are not new, and merely confirm the results of many previous studies. The contribution and novelty of the paper lie in the new methodology used to arrive at these results. In my view, the paper needs to be revised in terms of the Title (to emphasize the novelty of methodology) and in general presentation to emphasize the new measurement approach.

A recommendation for a new title might be "A New Approach for Quantifying Corneal Thickness Changes after Orthokeratology Contact Lens Wear". The focus of the abstract will need to change significantly to highlight the new approach and to recognise that the findings are very similar to those reported in previous papers.

Throughout the paper, the authors refer to and discuss changes that do not reach statistical significance as if they are "real" changes. This over-interpretation of effects must be avoided, as failure to reach statistical significance means that any apparent changes may simply be due to chance. The manuscript should be carefully revised throughout to avoid this error of interpretation.

I have also made some minor comments below.

Page 1, line 14: This is a prospective study.

Abstract, lines 43-44: Were the changes in paracentral epithelial thickness statistically significant? If so, please insert p-values. If not, this needs to be noted. I also find it difficult to reconcile the data presented here with the data in Table 2 for both epithelial and stromal thickness changes.

Line 69: Two of these references (Charm et al and Kakita et al) demonstrate only partial correction of myopia in patients with higher degrees of myopia, and the other reference simply refers to Charm and Kakita – indeed the sentence written here "Subsequent studies...up to -10.0 D" is directly copied from the Bullimore reference! This sentence needs to be reconsidered.

Line 69: "Yes"?? The following sentence needs to be rewritten. It is grammatically mangled, and fails to take into account the many papers that have attempted to explain the biomechanical effects of orthokeratology.

Line 108: Please decide whether the participants were patients or subjects (or participants) and be consistent with terminology.

Line 114: Define "unacceptable lens decentration".

Line 121: What lens material was used? It is likely to differ between the lens designs, and may not have a common Dk of 90.

There are many details about clinical care that are not included in the Methods, such as frequency of aftercares, lens solutions used, any lens parameter changes, any adverse effects etc. These details would be expected in a prospective clinical study such as this. This also further emphasizes that the main purpose of the paper was to apply the new methodology rather than to investigate the clinical impact of orthokeratology. Is it possible to refer to the clinical methodology previously published by some of the co-authors?

Lines 127 and following: It is not specifically stated if OCT images were obtained before and after one month of orthokeratology lens wear, and whether they were obtained between 9 AM and 11 AM. This is specifically stated for Pentacam data.

Lines 138-140: 52-7 does not equal 44 eyes - what happened to the other one?

Line 191: What was the visible width of the epithelial surface? I assume that this is what limited your analysis to a 5-mm chord?

Line 196: Is 10um correct? Or should line 190 explain that the "5 points" selected were on the posterior epithelial surface?

In Figure 3, please define nasal/temporal and superior/inferior directions.

Line 228: Is the negative value indicating the nasal or temporal meridian?

Line 242: The standard deviation values in the epithelium were significantly smaller than in the stroma. This would indicate more consistent epithelial thickness changes.

Line 281: Better to say "and after one month of overnight lens wear".

Line 288: Spelling error: "lens"

Line 286-288: The current study was over one month of lens wear. This is a very different period from 1 night or even 1 week, as reported in the other studies referenced here (except for ref 30, which was after 30 days of lens wear). The effects over more comparable periods of lens wear would be more appropriate to cite here.

Line 294: Redistribution of epithelial cells out from the corneal centre to the paracentral area, against the normal centripetal migration of epithelial cells from the limbus, seems an unlikely explanation.

Line 309 and following: It is unclear how soon after eye opening/lens removal the stromal thickness changes were measured. Given the relatively low Dk/t of orthokeratology lenses, is it possible that this increase in stromal thickness may be attributable to overnight hypoxia?

Line 315: The Alharbi paper did not find stromal thickness increases across the cornea, only in the paracentral area.

Line 324 and following: See my earlier comment about comparing results from the current study (one month duration) with similar duration studies.

Line 330-331: Refer to study by Yoon et al (OVS 2013), which examined posterior corneal curvature changes in orthokeratology.

Line 337: Were there 44 or 45 participants?

Line 348: There has been no previous mention about variations in follow-up time. This needs to be reported somewhere – if the range was large, then this would obviously introduce some variability.

References: The formatting of the references is very poor and haphazard, and this needs to be tidied up by the authors. In many cases essential reference information including authors names, journal name, and journal details are missing or incorrect. Refer to the journal's required style for reference formatting.

===PREPARING YOUR MANUSCRIPT===

===PREPARING YOUR REVISION IN SCHOLARONE===

Author's Response to Decision Letter for (RSOS-211108.R0)

See Appendix A.

RSOS-211108.R1 (Revision)

Review form: Reviewer 2

Is the manuscript scientifically sound in its present form?

Yes

Are the interpretations and conclusions justified by the results?

Yes

Is the language acceptable?

Yes

Do you have any ethical concerns with this paper?

No

Have you any concerns about statistical analyses in this paper?

No

Recommendation?

Accept with minor revision (please list in comments)

Comments to the Author(s)

The major revision of this paper to emphasize the primary purpose of the research (to apply a new approach for quantifying corneal variables), through revision of the Title, Abstract and some parts of the Discussion, has been successful and has overcome my major concern with the submitted version of the paper.

However, the revision of the document is quite rough, and there are still a number of issues that should be addressed in a minor revision. In particular I have concerns about the degree of intellectual focus in discussing and interpreting the study results – see my comments below about lines 293 and following. In some places I am not sure that the authors have really read the references – see my comments about page 74. There are still some instances of over-interpretation. There are many instances of poor grammar and misspellings. In some cases, although the authors have claimed to change the text, this has not been done. The reference list is improved but still a mishmash of formats.

Specific comments:

Previously Page 1, line 14, now line 391: Correction not made. This is a prospective study, not a perspective study.

Abstract: In general the changes in the Abstract better reflect the purpose and outcomes of the study. However, please note the following:

- Lines 36-37, and 41: These sentences should be reworded for clarity: “...were monitored before and after one month of Ortho-K lens wear”; and “...before and after 1 month wear of Ortho-K lenses.” Otherwise it could be interpreted that the second set of measurements were taken after Ortho-K lens wear had ceased for one month (i.e. recovery data).
- Line 43: “the epithelium”
- Line 46: Overinterpretation: the change in epithelial thickness in the superior and inferior midperiphery did not reach statistical significance. Thus the authors cannot interpret this to mean that thickening in fact took place in these regions.

- Line 47: “The stromal thickness”
- Line 54 and ff: The authors should mention somewhere here that these results are consistent with changes reported in previous papers on Ortho-K.

Line 65: I’m not sure that the word “naked” is appropriate here – maybe “unaided” or “uncorrected”.

Line 71: Reword this sentence: “High myopia has been partially reduced by Ortho-K lenses in some studies (7,8).”

Line 73: Reword and correct spelling error: “In several studies, epithelial thickness changes were recognized as...”

Line 74: Here in the list of references there is a “grab bag” of previous papers on Ortho-K, but only some of them actually talk about epithelial thickness changes. For example, ref 6 did not measure epithelial thickness. Ref 10 examined radii of curvature, not epithelial thickness. Ref 13 discusses myopia control with Ortho-K but does not discuss mechanisms of action of Ortho-K. Refs 2, 3, 11 and 13 are review papers rather than primary references. More thoughtful choice of appropriate references is recommended.

Line 128: The Boston Equalens II material has a Dk of 85, not 90. Please correct this error.

Line 146: See my earlier comment in the Abstract. Reword as: “...at baseline and after one month of Ortho-K lens wear...”. This comment also applies at other places in the paper – please review.

Line 168: Reword: “...found adequate in preliminary study trials)...”

Line 234: Correct spelling error: “...visible width of the epithelial surface on OCT images.”

Line 237: “Tables 1 and 2”

Line 288 (was line 281): the text has not been changed as suggested – please change (see my earlier comments).

Line 293 and following: Although it is good that the discussion now compares the results from this study with those of comparable duration, the authors have not thought carefully about their comments! The results of this study after 1 month of lens wear, showing about 15 microns of epithelial thinning, is clearly NOT consistent with Zhang et al’s finding of 4.5 microns of thinning over a similar period. They are more consistent with the 18 microns of thinning reported in ref 33 (Reinstein et al), but ref 33 is a single case study. A better comparison might be with the Alharbi et al paper (ref 36), which reported epithelial thinning of just over 15 microns (see their Fig 2) after 1 month of Ortho-K lens wear.

- The use of raw numbers and percentages in this paragraph is confusing – try to be more consistent.
- It is not meaningful to quote micron changes to two decimal places.
- Some comment about the variability of outcomes from different studies of similar duration is warranted.

Please review this section and rewrite with more thoughtful consideration!

Line 304: Rewrite: “...limbus could be another reason.”

Line 324-6: This sentence needs grammatical work. I suggest “...of central stromal thinning and thickening in the paracentral area...although these did not reach statistical significance...”

Line 327: “months”

Lines 362-4: This sentence needs grammatical work. I suggest “Finally, variations in the follow-up time after commencing lens wear, due to booking restrictions and participants’ personal issues (34 ± 6 days, range: 27 - 43) may have caused some variability in changes in thickness and radius of curvature.”

- Note that there is no need to give the day period to 2 decimal places.

Line 368: Spelling error: “flattened”

Line 398: Delete “...would be excluded”.

Line 400: Change “subjects” to “participants”.

References: Although the formatting of the references has improved, there are still some inconsistencies. For example, some of the paper titles are capitalized while others are not. Many journal names are not appropriately abbreviated.

Decision letter (RSOS-211108.R1)

Dear Ms Ran

On behalf of the Editors, we are pleased to inform you that your Manuscript RSOS-211108.R1 "A New Approach for Quantifying Epithelial and Stromal Thickness Changes after Orthokeratology Contact Lens Wear" has been accepted for publication in Royal Society Open Science subject to minor revision in accordance with the referees' reports. Please find the referees' comments along with any feedback from the Editors below my signature.

Please submit your revised manuscript and required files (see below) no later than 7 days from today's (ie 23-Nov-2021) date. Note: the ScholarOne system will 'lock' if submission of the revision is attempted 7 or more days after the deadline. If you do not think you will be able to meet this deadline please contact the editorial office immediately.

on behalf of Dr Adil Al-Mayah (Associate Editor) and R. Kerry Rowe (Subject Editor)
 openscience@royalsociety.org

Associate Editor Comments to Author (Dr Adil Al-Mayah):

Significant modifications have been made to the paper that will improve its potential impact. However, given the level of these changes, the reviewer has proposed some important modifications to further improve the paper and its presentation.

Reviewer comments to Author:

Reviewer: 2

Comments to the Author(s)

The major revision of this paper to emphasize the primary purpose of the research (to apply a new approach for quantifying corneal variables), through revision of the Title, Abstract and some parts of the Discussion, has been successful and has overcome my major concern with the submitted version of the paper.

However, the revision of the document is quite rough, and there are still a number of issues that should be addressed in a minor revision. In particular I have concerns about the degree of intellectual focus in discussing and interpreting the study results – see my comments below about lines 293 and following. In some places I am not sure that the authors have really read the references – see my comments about page 74. There are still some instances of over-interpretation. There are many instances of poor grammar and misspellings. In some cases, although the authors have claimed to change the text, this has not been done. The reference list is improved but still a mishmash of formats.

Specific comments:

Previously Page 1, line 14, now line 391: Correction not made. This is a prospective study, not a perspective study.

Abstract: In general the changes in the Abstract better reflect the purpose and outcomes of the study. However, please note the following:

- Lines 36-37, and 41: These sentences should be reworded for clarity: "...were monitored before and after one month of Ortho-K lens wear"; and "...before and after 1 month wear of Ortho-K lenses." Otherwise it could be interpreted that the second set of measurements were taken after Ortho-K lens wear had ceased for one month (i.e. recovery data).
- Line 43: "the epithelium"
- Line 46: Overinterpretation: the change in epithelial thickness in the superior and inferior midperiphery did not reach statistical significance. Thus the authors cannot interpret this to mean that thickening in fact took place in these regions.
- Line 47: "The stromal thickness"
- Line 54 and ff: The authors should mention somewhere here that these results are consistent with changes reported in previous papers on Ortho-K.

Line 65: I'm not sure that the word "naked" is appropriate here - maybe "unaided" or "uncorrected".

Line 71: Reword this sentence: "High myopia has been partially reduced by Ortho-K lenses in some studies (7,8)."

Line 73: Reword and correct spelling error: "In several studies, epithelial thickness changes were recognized as..."

Line 74: Here in the list of references there is a "grab bag" of previous papers on Ortho-K, but only some of them actually talk about epithelial thickness changes. For example, ref 6 did not measure epithelial thickness. Ref 10 examined radii of curvature, not epithelial thickness. Ref 13 discusses myopia control with Ortho-K but does not discuss mechanisms of action of Ortho-K. Refs 2, 3, 11 and 13 are review papers rather than primary references. More thoughtful choice of appropriate references is recommended.

Line 128: The Boston Equalens II material has a Dk of 85, not 90. Please correct this error.

Line 146: See my earlier comment in the Abstract. Reword as: "...at baseline and after one month of Ortho-K lens wear...". This comment also applies at other places in the paper - please review.

Line 168: Reword: "...found adequate in preliminary study trials)..."

Line 234: Correct spelling error: "...visible width of the epithelial surface on OCT images."

Line 237: "Tables 1 and 2"

Line 288 (was line 281): the text has not been changed as suggested - please change (see my earlier comments).

Line 293 and following: Although it is good that the discussion now compares the results from this study with those of comparable duration, the authors have not thought carefully about their comments! The results of this study after 1 month of lens wear, showing about 15 microns of epithelial thinning, is clearly NOT consistent with Zhang et al's finding of 4.5 microns of thinning over a similar period. They are more consistent with the 18 microns of thinning reported in ref 33 (Reinstein et al), but ref 33 is a single case study. A better comparison might be with the Alharbi et al paper (ref 36), which reported epithelial thinning of just over 15 microns (see their Fig 2) after 1 month of Ortho-K lens wear.

- The use of raw numbers and percentages in this paragraph is confusing - try to be more consistent.
- It is not meaningful to quote micron changes to two decimal places.
- Some comment about the variability of outcomes from different studies of similar duration is warranted.

Please review this section and rewrite with more thoughtful consideration!

Line 304: Rewrite: "...limbus could be another reason."

Line 324-6: This sentence needs grammatical work. I suggest "...of central stromal thinning and thickening in the paracentral area...although these did not reach statistical significance..."

Line 327: "months"

Lines 362-4: This sentence needs grammatical work. I suggest “Finally, variations in the follow-up time after commencing lens wear, due to booking restrictions and participants’ personal issues (34 ± 6 days, range: 27 - 43) may have caused some variability in changes in thickness and radius of curvature.”

- Note that there is no need to give the day period to 2 decimal places.

Line 368: Spelling error: “flattened”

Line 398: Delete “...would be excluded”.

Line 400: Change “subjects” to “participants”.

References: Although the formatting of the references has improved, there are still some inconsistencies. For example, some of the paper titles are capitalized while others are not. Many journal names are not appropriately abbreviated.

===PREPARING YOUR MANUSCRIPT===

one version should clearly identify all the changes that have been made (for instance, in coloured highlight, in bold text, or tracked changes);

===PREPARING YOUR REVISION IN SCHOLARONE===

To revise your manuscript, log into <https://mc.manuscriptcentral.com/rsos> and enter your Author Centre - this may be accessed by clicking on "Author" in the dark toolbar at the top of the

page (just below the journal name). You will find your manuscript listed under "Manuscripts with Decisions". Under "Actions", click on "Create a Revision".

-- Ensure that your data access statement meets the requirements at <https://royalsociety.org/journals/authors/author-guidelines/#data>.

You should ensure that you cite the dataset in your reference list. If you have deposited data etc in the Dryad repository, please only include the 'For publication' link at this stage. You should remove the 'For review' link.

-- If you are requesting an article processing charge waiver, you must select the relevant waiver option (if requesting a discretionary waiver, the form should have been uploaded, see 'File upload' above).

-- If you have uploaded any electronic supplementary (ESM) files, please ensure you follow the guidance at <https://royalsociety.org/journals/authors/author-guidelines/#supplementary-material> to include a suitable title and informative caption. An example of appropriate titling and captioning may be found at https://figshare.com/articles/Table_S2_from_Is_there_a_trade-off_between_peak_performance_and_performance_breadth_across_temperatures_for_aerobic_scope_in_teleost_fishes_/3843624.

At the 'Review & submit' step, you must view the PDF proof of the manuscript before you will be able to submit the revision. Note: if any parts of the electronic submission form have not been

completed, these will be noted by red message boxes - you will need to resolve these errors before you can submit the revision.

Author's Response to Decision Letter for (RSOS-211108.R1)

See Appendix B.

Decision letter (RSOS-211108.R2)

Dear Ms Ran,

I am pleased to inform you that your manuscript entitled "A New Approach for Quantifying Epithelial and Stromal Thickness Changes after Orthokeratology Contact Lens Wear" is now accepted for publication in Royal Society Open Science.

Kind regards,
Royal Society Open Science Editorial Office
Royal Society Open Science

on behalf of Dr Adil Al-Mayah (Associate Editor) and R. Kerry Rowe (Subject Editor)
openscience@royalsociety.org

Appendix A

Manuscript ID RSOS-211108

Dear Dr Adil Al-Mayah,

Thank you for giving us the opportunity to submit a revised draft of the manuscript “A New Approach for Quantifying Epithelial and Stromal Thickness Changes after Orthokeratology Contact Lens Wear” for publication in the Journal of Royal Society Open Science. We appreciate the time and effort that you and the reviewers dedicated to providing feedback on our manuscript and are grateful for the insightful comments on and valuable improvements to our paper. We have incorporated most of the suggestions made by the reviewers. Those changes are highlighted within the manuscript.

Please see below, in blue, for a point-by-point response to the reviewers’ comments and concerns.

Comments from Associate Editor Comments (Dr Adil Al-Mayah):

The paper presents an interesting study with great potential contribution to the field. The novelty of the method to obtain the reported results is well recognized. However, as raised by one reviewer, some concerns regarding the novelty and validity of the results need to be address.

Authors response: Thanks for your opinion. We have made modifications based on these concerns.

Comments from Reviewer 1

This is a solid piece of work that clarifies the mechanisms underlying the temporary changes induced by orthokeratology. My comments are minor.

Authors response: Thank you!

Lines 138-140: numbers don't add up

Authors response: Thank for pointing out this mistake. We have corrected the text to: “Therefore, a total of 45 eyes were included in the analysis.”

Line 160: "curvature" is the inverse of radius. I suggest a different word be used.

Authors response: Thank you for pointing this out. We agree with the reviewer, therefore the term “radius of curvature” was used in the revised version of the manuscript.

Comments from Reviewer 2

In this study, 52 patients wore orthokeratology lenses in an overnight wear modality for up to one month. A custom-built MATLAB code was used for semi-automatic segmentation of corneal OCT images in the two principal meridians out to a 5-mm chord, and changes in epithelial and stromal thickness after lens wear were measured. Central epithelial thinning was found to explain most of the refractive effect of orthokeratology, and some paracentral thickening was also noted. No significant changes in anterior or posterior stromal radius of curvature were found.

Overall, this is an interesting study, but presents very similar results as have been published many times before over the last 20 years in terms of changes in corneal epithelial and stromal thickness, and radii of curvature of the various corneal layers resulting from orthokeratology. In other words, the findings are not new, and merely confirm the results of many previous studies. The contribution and novelty of the paper lie in the new methodology used to arrive at these results. In my view, the paper needs to be revised in terms of the Title (to emphasize the novelty of methodology) and in general presentation to emphasize the new measurement approach.

A recommendation for a new title might be “A New Approach for Quantifying Corneal Thickness Changes after Orthokeratology Contact Lens Wear”. The focus of the abstract will need to change significantly to highlight the new approach and to recognise that the findings are very similar to those reported in previous papers.

Throughout the paper, the Authors refer to and discuss changes that do not reach statistical significance as if they are “real” changes. This over-interpretation of effects must be avoided, as failure to reach statistical significance means that any apparent changes may simply be due to chance. The manuscript should be carefully revised throughout to avoid this error of interpretation.

Authors response: Thank you for all the opinions and suggestions. As suggested, we changed the title to “A New Approach for Quantifying Epithelial and Stromal Thickness Changes after Orthokeratology Contact Lens Wear”; and changed the abstract to emphasise the novelty of the processing method in the purpose and conclusion sections. We also revised the results sections to avoid over-interpretation, such as the stromal thickness changes which did not reach statistical significance.

Here are parts of the abstract after making these changes:

“ Purpose: To develop an automatic segmentation approach of optical coherence tomography (OCT) images and to investigate the changes in epithelial and stromal thickness profile and radius of curvature after use of orthokeratology (Ortho-K) contact lenses.

Conclusions: This study introduced a new method to automatically detect the anterior corneal surface, the epithelial posterior surface, and the posterior corneal surface in OCT scans.”

The epithelium and stromal thickness results section has changed into:

“The thickness changes in the epithelium, stroma and the whole cornea were evaluated within the central area (0 – 2 mm diameter) and the paracentral area (2 – 5 mm diameter) in Table 1 and Figures 3 and 4. Overall, the epithelial thickness decreased by $12.8 \pm 6.0 \mu\text{m}$ (23.8% on average, $p < 0.01$) in the central area, which was larger than the decreases in the paracentral area ($2.4 \pm 5.9 \mu\text{m}$, 4.5% on average, $p = 0.031$ along the horizontal meridian and $2.2 \pm 7.1 \mu\text{m}$, 4.2% on average, $p = 0.086$ along the vertical meridian). In the central area, the epithelium experienced thinning of up to $14.8 \mu\text{m}$ at the centre in the vertical scans ($p < 0.01$, Figure 4), and up to $15.0 \mu\text{m}$ at -0.2mm away from the centre in the horizontal scans ($p < 0.01$, Figure 4). The significance of the thickness changes was close to 0 in the area from $X = -1.5 \text{ mm}$ to $X = 1.5 \text{ mm}$ in both meridians, then the changes became insignificant from a radius of 1.7 mm onwards.”

“In these results, the mean changes in stromal thickness were smaller than corresponding changes in epithelial thickness ($p < 0.05$), while the standard deviation values in the epithelium were significantly smaller than in the stroma ($12.8 \pm 6.0 \mu\text{m}$ vs $4.8 \pm 16.1 \mu\text{m}$, $p < 0.01$), indicating much

more consistent changes in epithelial thickness. The stromal thickness changes were significant at X = -0.5mm to X = 0 mm in the horizontal meridian (all p < 0.05), elsewhere the changes were not significant.”

Page 1, line 14: This is a prospective study.

Authors response: This has been corrected in the revised version of the manuscript.

Abstract, lines 43-44: Were the changes in paracentral epithelial thickness statistically significant? If so, please insert p-values. If not, this needs to be noted. I also find it difficult to reconcile the data presented here with the data in Table 2 for both epithelial and stromal thickness changes.

Authors response: As suggested by the reviewer, the p-values were included. Now we have” In the paracentral area (2–5 mm diameter), the epithelium thinned nasally and temporally (by $2.4\pm 5.9\ \mu\text{m}$, 4.5% on average, p = 0.031), and thickened superiorly and inferiorly (by $2.2\pm 7.1\ \mu\text{m}$, 4.2% on average, p = 0.086).”

The data in Table 2 contained the mean and standard deviation of the thickness measured pre and post-OrthoK lens wear, while the data mentioned in line 43-44 were the thickness changes calculated from Table 2, and included in the “Results” section as below:

“Overall, the epithelial thickness decreased by $12.8\pm 6.0\ \mu\text{m}$ (23.8% on average, p < 0.01) in the central area, which was larger than the thickness changes in the paracentral area (decreased $2.4\pm 5.9\ \mu\text{m}$, 4.5% on average, p = 0.031 along the horizontal meridian and increased $2.2\pm 7.1\ \mu\text{m}$, 4.2% on average, p = 0.086 along the vertical meridian).” And “The thickness increase was $4.8\pm 16.1\ \mu\text{m}$ (p < 0.01) in the central area, while in the paracentral area, the increase was $3.6\pm 20.0\ \mu\text{m}$ (p = 0.272) along the vertical meridian, and $2.4\pm 17.2\ \mu\text{m}$ (p = 0.418) along the horizontal meridian.”

Line 69: Two of these references (Charm et al and Kakita et al) demonstrate only partial correction of myopia in patients with higher degrees of myopia, and the other reference simply refers to Charm and Kakita – indeed the sentence written here “Subsequent studies...up to -10.0 D” is directly copied from the Bullimore reference! This sentence needs to be reconsidered.

Authors response: Thanks for pointing this out. We changed this into:” High myopia has been partially reduced by Ortho-K lenses in the following studies ^[7,8].”

Line 69: “Yes”?? The following sentence needs to be rewritten. It is grammatically mangled, and fails to take into account the many papers that have attempted to explain the biomechanical effects of orthokeratology.

Authors response: Thanks for pointing this out. We have “Yet, despite interest in Ortho-K lenses, the mechanism by which they interact with the cornea is still not fully understood. In several studies, the epithelial thickness changes were recognized as the main cause of corneal reshaping ^[2, 3, 6, 9-16].”

Line 108: Please decide whether the participants were patients or subjects (or participants) and be consistent with terminology.

Authors response: As suggested by the reviewer, we now use “participants” throughout the whole manuscript.

Line 114: Define “unacceptable lens decentration”.

Authors response: Thanks for pointing this out. We have now defined unacceptable decentration in the new version: "... or unacceptable lens decentration (over 1mm) after regular wear were excluded".

Line 121: What lens material was used? It is likely to differ between the lens designs, and may not have a common Dk of 90.

Authors response: We have used 3 brands of OrthoK lenses in the study. Details including the Dk values have been included in the manuscript in Page 5: "The Euclid lenses, manufactured with Boston Equalens II material, had Dk of $90 \times 10^{-11}(\text{cm}^2/\text{sec})$ ($\text{mLO}_2/\text{mL} \times \text{mmHg}$), while the Paragon CRT lenses were made of Paragon HDS 100 Paflucocon D material with Dk of $100 \times 10^{-11}(\text{cm}^2/\text{sec})$ ($\text{mLO}_2/\text{mL} \times \text{mmHg}$). Finally, the Dreamlite lenses were manufactured with the Boston XO hexafocon-A material that had Dk of $100 \times 10^{-11}(\text{cm}^2/\text{sec})(\text{mLO}_2/\text{mL} \times \text{mmHg})$."

There are many details about clinical care that are not included in the Methods, such as frequency of aftercares, lens solutions used, any lens parameter changes, any adverse effects etc. These details would be expected in a prospective clinical study such as this. This also further emphasizes that the main purpose of the paper was to apply the new methodology rather than to investigate the clinical impact of orthokeratology. Is it possible to refer to the clinical methodology previously published by some of the co-Authors?

Authors response: In response to your comment, we have added reference to an earlier study by the clinical Authors in which the information listed in your comment is included. Here is the new text (Page 4):

Authors "The inclusion criteria included age between 8 and 18 years, best corrected visual acuity (BCVA) greater than 20/25, spherical equivalent between -1.00 D and -6.00 D, and intraocular pressure between 11 and 21mmHg. Participants with history of Ortho-K lens wear, contact lens contra-indications, related eye systemic diseases, uncorrected visual acuity below 20/25, or unacceptable lens decentration (over 1mm) after regular wear were excluded. These inclusion and exclusion criteria, along with details of the clinical care adopted, follow those presented in our earlier clinical study ^[29]."

Lines 127 and following: It is not specifically stated if OCT images were obtained before and after one month of orthokeratology lens wear, and whether they were obtained between 9 AM and 11 AM. This is specifically stated for Pentacam data.

Authors response: As suggested by the reviewer, we rewrote this part (Page 5) to make this point clear: "All OCT and Pentacam measurements were performed by an experienced operator (FJ) at baseline and one-month post Ortho-K lens wear between 9 AM and 11 AM to minimise the diurnal variation. The measurements were repeated until an examination quality of "OK" was obtained."

Lines 138-140: 52-7 does not equal 44 eyes – what happened to the other one?

Authors response: Thank you for noting this mistake. We have modified the text to: "Therefore, a total of 45 eyes were included in the analysis."

Line 191: What was the visible width of the epithelial surface? I assume that this is what limited your analysis to a 5-mm chord?

Authors response: The visible width of the epithelium surface varied between individuals, but was around 5 mm in most cases. This was why we chose 5 mm to be the analysis limit. This point was explained in the text: (Page 9) “ This study focused on the corneal area within 5mm diameter because of the limitation in visible width of the epithelium surface on OCT images.”

Line 196: Is 10um correct? Or should line 190 explain that the “5 points” selected were on the posterior epithelial surface?

Authors response: Yes, 10µm is correct, and was used by the custom-built code. As suggested by the reviewer, we have changed the original text from: “At least 5 points on the epithelium surface” to: “At least 5 points on the epithelium posterior surface”.

In Figure 3, please define nasal/temporal and superior/inferior directions.

Authors response: As suggested, we have added the “superior/nasal” and “inferior/temporal” notation to Figure 3.

Figure 3 Mean epithelial thickness (a) before and (b) after Ortho-K lens wear

Line 228: Is the negative value indicating the nasal or temporal meridian?

Authors response: The negative value indicates the inferior or the temporal meridian. We have added the meridians in Figure 3 to clarify the directions as mentioned above.

Line 242: The standard deviation values in the epithelium were significantly smaller than in the stroma. This would indicate more consistent epithelial thickness changes.

Authors response: Thank you for noting this point. We have now noted (in Page 10): “... while the standard deviation values in the epithelium were significantly smaller than in the stroma ($12.8 \pm 6.0 \mu\text{m}$ vs $4.8 \pm 16.1 \mu\text{m}$, $p < 0.01$), indicating much more consistent changes in epithelial thickness”.

Line 281: Better to say “and after one month of overnight lens wear”.

Authors response: Thank you. We have changed the text as suggested:” the stroma and the total cornea in Ortho-K lens wearers before and after one month of overnight lens wear”.

Line 288: Spelling error: “lens”

Authors response: Thank you for pointing out this mistake. It has been corrected.

Line 286-288: The current study was over one month of lens wear. This is a very different period

from 1 night or even 1 week, as reported in the other studies referenced here (except for ref 30, which was after 30 days of lens wear). The effects over more comparable periods of lens wear would be more appropriate to cite here.

Authors response: Thanks for your suggestion. We have adjusted the order of earlier studies to bring those with a 30 day follow-up first. As there were only few studies, in which the follow up was for 1 month, we included those with shorter follow-up but noted the difference. This point is now explained in Page 12:

“These results are consistent with earlier reports in which the central epithelial thickness decreased from $52.04 \pm 2.35 \mu\text{m}$ to $47.53 \pm 3.44 \mu\text{m}$ ^[15], and by up to $18 \mu\text{m}$ following 30 days of lens wear ^[34]. Other studies reported epithelium thickness reductions after 1 day of lens wear of 5.4% ^[25], $5.1 \pm 4.5\%$ ^[35], $4.63 \pm 2.72 \mu\text{m}$ ^[36], and $8.7 \pm 4.8 \mu\text{m}$ ^[37]. Other studies adopted a 1 week follow up and reported epithelial thickness reductions of 14.3% ^[25], $10.62 \pm 4.25 \mu\text{m}$ ^[36], and $6.13 \pm 1.67 \mu\text{m}$ ^[15].”

Line 294: Redistribution of epithelial cells out from the corneal centre to the paracentral area, against the normal centripetal migration of epithelial cells from the limbus, seems an unlikely explanation.

Authors response: Thanks for pointing this out. We have modified this part to “...and remodelling of the epithelium due to interference by the contact pressure with the normal centripetal migration of epithelial cells from the limbus could be the another reason.”

Line 309 and following: It is unclear how soon after eye opening/lens removal the stromal thickness changes were measured. Given the relatively low Dk/t of orthokeratology lenses, is it possible that this increase in stromal thickness may be attributable to overnight hypoxia?

Authors response: Thanks for sharing this idea. We agree that oedema could be the reason behind the increases in stromal thickness, and have included reference to this possibility in the manuscripts Page 13): “while oedema by overnight hypoxia is a likely reason for central thickening”.

Line 315: The Alharbi paper did not find stromal thickness increases across the cornea, only in the paracentral area.

Authors response: Thanks for noting this point. The text has been changed to “and another by Alharbi and Swarbrick showing stromal thickness increases in the paracentral area over three month of lens wear ^[37].”

Line 324 and following: See my earlier comment about comparing results from the current study (one month duration) with similar duration studies.

Authors response: Thanks for your suggestion. As mentioned above, we have responded to this point and the new text, included in Page 13, is: “Stromal thickness experienced small, but significant, increases in the central area with a large scatter in values. These findings are compatible with Reinstein’s results ^[34], in which the stromal thickness was $5 \mu\text{m}$ larger, on average, after two weeks of orthokeratology within the central 3-mm diameter area. In the paracentral area, the stromal thickness experienced only small and non-significant variations after lens wear. These results are similar, to some extent, to a report by Nieto-Bona et al. ^[23] of stromal thinning in central and thickening in paracentral area after one month of lens wear but not significant, and another by

Alharbi and Swarbrick showing stromal thickness increases in the paracentral area over three month of lens wear ^[37].”

Line 330-331: Refer to study by Yoon et al (OVS 2013), which examined posterior corneal curvature changes in orthokeratology.

Authors response: As suggested, we have added this reference in Page 13 as follows:” Yoon et al. reported no significant changes in either posterior corneal apical radius or asphericity after 14 days of lens wear ^[41].”

Line 337: Were there 44 or 45 participants?

Authors response: We have corrected the mistake in the number of participants as indicated earlier, and the new text is: “Therefore, a total of 45 eyes were included in the analysis.”

Line 348: There has been no previous mention about variations in follow-up time. This needs to be reported somewhere – if the range was large, then this would obviously introduce some variability.

Authors response: In this study, participants were asked to arrange an appointment 30 days after wearing the lens, but as it was difficult to stick to this target due to booking restrictions and participants’ personal issues, there was some variation in the actual follow-up periods. This point was explained in the text in Page 14: “Finally, the variations in the follow-up time post lens wear – due to booking restrictions and participants’ personal issues (34.04 ± 5.80 days, range: 27 - 41) may have caused differences in the thickness and radius of curvature.”

References: The formatting of the references is very poor and haphazard, and this needs to be tidied up by the Authors. In many cases essential reference information including Authors names, journal name, and journal details are missing or incorrect. Refer to the journal’s required style for reference formatting.

Authors response: As suggested by the reviewer, we checked and reorganised the references in the correct format.

We look forward to hearing from you in due course regarding our submission and to respond to any further questions and comments you may have.

Sincerely,
Ziying Ran

Appendix B

Manuscript ID RSOS-211108

Dear Dr Adil Al-Mayah,

Thank you for reviewing our manuscript “A New Approach for Quantifying Epithelial and Stromal Thickness Changes after Orthokeratology Contact Lens Wear” for publication in the Journal of Royal Society Open Science. We appreciate the time and effort that you and the reviewers dedicated to our manuscript and are grateful for the insightful comments on and valuable improvements to our paper. We have incorporated all of the suggestions made by the reviewers. Those changes are highlighted within the manuscript.

Please see below, in blue, a point-by-point response to the reviewers’ comments.

Comments from Associate Editor Comments (Dr Adil Al-Mayah):

Significant modifications have been made to the paper that will improve its potential impact. However, given the level of these changes, the reviewer has proposed some important modifications to further improve the paper and its presentation.

Authors response: Thanks. We have made modifications based on the reviewers’ suggestions.

Comments from Reviewer 2

The major revision of this paper to emphasize the primary purpose of the research (to apply a new approach for quantifying corneal variables), through revision of the Title, Abstract and some parts of the Discussion, has been successful and has overcome my major concern with the submitted version of the paper.

However, the revision of the document is quite rough, and there are still a number of issues that should be addressed in a minor revision. In particular I have concerns about the degree of intellectual focus in discussing and interpreting the study results – see my comments below about lines 293 and following. In some places I am not sure that the authors have really read the references – see my comments about page 74. There are still some instances of over-interpretation. There are many instances of poor grammar and misspellings. In some cases, although the authors have claimed to change the text, this has not been done. The reference list is improved but still a mishmash of formats.

Authors response: Many thanks for the comments made to improve our manuscript. Modifications have been made based on these comments.

Specific comments:

Previously Page 1, line 14, now line 391: Correction not made. This is a prospective study, not a perspective study.

Authors response: This has been corrected in the revised manuscript.

Abstract: In general, the changes in the Abstract better reflect the purpose and outcomes of the study. However, please note the following:

- Lines 36-37, and 41: These sentences should be reworded for clarity: "...were monitored before and after one month of Ortho-K lens wear"; and "...before and after 1 month wear of Ortho-K lenses." Otherwise, it could be interpreted that the second set of measurements were taken after Ortho-K lens wear had ceased for one month (i.e. recovery data).

Authors response: Thanks for pointing this out. Now we have: "A total of 45 right eyes from 52 participants were monitored before and after one month of uninterrupted overnight Ortho-K lens wear." And "... changes in epithelial thickness, stromal thickness, corneal and stromal profiles and radii of curvature before, and after one month of, uninterrupted overnight wear of Ortho-K lenses."

- Line 43: "the epithelium"

Authors response: This has been modified to "the epithelium thinned by $12.8 \pm 6.0 \mu\text{m}$."

- Line 46: Overinterpretation: the change in epithelial thickness in the superior and inferior midperiphery did not reach statistical significance. Thus, the authors cannot interpret this to mean that thickening in fact took place in these regions.

Authors response: Thanks for pointing this out, we now have "In the paracentral area (2–5 mm diameter), the epithelium thinned nasally and temporally (by $2.4 \pm 5.9 \mu\text{m}$, 4.5% on average, $p = 0.031$)."

- Line 47: "The stromal thickness"

Authors response: This has been modified to "The stroma thickness increased in the central area (by $4.8 \pm 16.1 \mu\text{m}$, $p = 0.005$)."

- Line 54 and ff: The authors should mention somewhere here that these results are consistent with changes reported in previous papers on Ortho-K.

Authors response: As suggested by the reviewer, we now have" ... The anterior corneal surface became flattered while the anterior and posterior surfaces of the stroma did not undergo significant changes. The results are consistent with the changes reported in previous studies."

Line 65: I'm not sure that the word "naked" is appropriate here – maybe "unaided" or "uncorrected".

Authors response: Thanks for the suggestion, we have changed the word to "unaided".

Line 71: Reword this sentence: "High myopia has been partially reduced by Ortho-K lenses in some studies (7,8)."

Authors response: As suggested by the reviewer, this has been modified to "High myopia has been partially reduced by Ortho-K lenses in some studies (7, 8)."

Line 73: Reword and correct spelling error: "In several studies, epithelial thickness changes were recognized as..."

Authors response: As suggested by the reviewer, this has been changed to "In several studies, epithelial thickness changes were recognized as the main cause of corneal reshaping."

Line 74: Here in the list of references there is a "grab bag" of previous papers on Ortho-K, but only some of them actually talk about epithelial thickness changes. For example, ref 6 did not measure epithelial thickness. Ref 10 examined radii of curvature, not epithelial thickness. Ref 13 discusses myopia control with Ortho-K but does not discuss mechanisms of action of Ortho-K. Refs 2, 3, 11 and 13 are review papers rather than primary references. More thoughtful choice of appropriate references is recommended.

Authors response: Thanks for pointing this out. We have reorganised the references in this part to focus more on epithelial thickness changes and exclude those which did not have a direct connection. We now have: "In several studies, epithelial thickness changes were recognized as the main cause of corneal reshaping (9-16)."

Line 128: The Boston Equalens II material has a Dk of 85, not 90. Please correct this error.

Authors response: Thanks for noting this mistake, the text has been changed to "The Euclid lenses, manufactured with Boston Equalens II material, had Dk of $85 \times 10^{-11}(\text{cm}^2/\text{sec})$ ($\text{mLO}_2/\text{mL} \times \text{mmHg}$)".

Line 146: See my earlier comment in the Abstract. Reword as: "...at baseline and after one month of Ortho-K lens wear...". This comment also applies at other places in the paper – please review.

Authors response: This has been corrected to "All OCT and Pentacam HR measurements were performed by an experienced operator (FJ) at baseline and after one-month of Ortho-K lens wear. The measurements were taken between 9 AM and 11 AM to minimise ...".

Line 168: Reword: "...found adequate in preliminary study trials) ..."

Authors response: Thanks for pointing this out, this has been modified to "with a value of 0.01 mm, which was found adequate in our preliminary study trials".

Line 234: Correct spelling error: "...visible width of the epithelial surface on OCT images."

Authors response: Thanks for pointing out the mistake, this has been corrected.

Line 237: "Tables 1 and 2"

Authors response: As suggested, the text has been changed to: "... within the central area (0 – 2 mm diameter) and the paracentral area (2 – 5 mm diameter) in Tables 1 and 2, and Figures 3 and 4."

Line 288 (was line 281): the text has not been changed as suggested – please change (see my earlier comments).

Authors response: This part has been changed to "the stroma and the total cornea in Ortho-K lens wearers before, and after one month of, overnight lens wear."

Line 293 and following: Although it is good that the discussion now compares the results from this study with those of comparable duration, the authors have not thought carefully about their comments! The results of this study after 1 month of lens wear, showing about 15 microns of epithelial thinning, is clearly NOT consistent with Zhang et al's finding of 4.5 microns of thinning over a similar period. They are more consistent with the 18 microns of thinning reported in ref 33 (Reinstein et al), but ref 33 is a single case study. A better comparison might be with the Alharbi et al paper (ref 36), which reported epithelial thinning of just over 15 microns (see their Fig 2) after 1 month of Ortho-K lens wear.

- The use of raw numbers and percentages in this paragraph is confusing – try to be more consistent.
- It is not meaningful to quote micron changes to two decimal places.
- Some comment about the variability of outcomes from different studies of similar duration is warranted.

Please review this section and rewrite with more thoughtful consideration!

Authors response: Thanks for the suggestion, we agree that Alharbi's results are more consistent with our results than others. The two decimals were reduced to one in the text. And now we have "The OCT image analysis indicated significant reductions in central epithelial thickness by $14.8 \pm 5.9 \mu\text{m}$ (or 26% of initial thickness) attributable to Ortho-K treatment ($p < 0.01$). These results are consistent with earlier reports in which the central epithelial thickness decreased over 30 days of lens wear by $15.8 \pm 3.3 \mu\text{m}$ in 18 eyes (12), and by $18 \mu\text{m}$ in one eye (32). Other studies reported epithelium thickness reduction after one week of lens wear of $10.6 \pm 4.2 \mu\text{m}$ (10), and $6.1 \pm 1.6 \mu\text{m}$ (15). Others adopted a one day follow up and reported epithelial thickness reductions of $4.6 \pm 2.7 \mu\text{m}$ (10), and $8.7 \pm 4.8 \mu\text{m}$ (12)."

Line 304: Rewrite: "...limbus could be another reason."

Authors response: Thanks for pointing out. This has been corrected.

Line 324-6: This sentence needs grammatical work. I suggest "...of central stromal thinning and thickening in the paracentral area...although these did not reach statistical significance..."

Authors response: As suggested by the reviewer, we have "...to a report by Nieto-Bona et al.(23) of central stromal thinning and thickening in paracentral area after one month of lens wear although these did not reach statistical significance...".

Line 327: "months"

Authors response: This mistake has been corrected.

Lines 362-4: This sentence needs grammatical work. I suggest "Finally, variations in the follow-up time after commencing lens wear, due to booking restrictions and participants' personal issues (34 ± 6 days, range: 27 - 43) may have caused some variability in changes in thickness and radius of curvature."

- Note that there is no need to give the day period to 2 decimal places.

Authors response: As suggested by the reviewer, we modified this part to "Finally, variations in the follow-up time after commencing lens wear, due to booking restrictions and participants' personal issues (34 ± 6 days, range: 27 – 43) may have caused some variability in changes in thickness and

radius of curvature.”

Line 368: Spelling error: “flattened”

Authors response: This mistake has been corrected to “The anterior corneal surface became flattened...”

Line 398: Delete “...would be excluded”.

Authors response: This part of the text is now:

“The inclusion criteria were as follows: 8 to 18 years of age, spherical equivalent from -1.00 D to -6.00 D, best-corrected visual acuity greater than 20/25, intraocular pressure within normal limits (11~21mmHg), no history of Ortho-K lens wear, no contact lens contraindications or related eye and systemic diseases. Patients who had uncorrected visual acuity less than 20/25 or unacceptable lens decentration after regular wear for one month had been excluded from the study.”

Line 400: Change “subjects” to “participants”.

Authors response: This has been changed to “All the examinations were conducted after the participants and their guardians fully understood and signed informed consent.”

References: Although the formatting of the references has improved, there are still some inconsistencies. For example, some of the paper titles are capitalized while others are not. Many journal names are not appropriately abbreviated.

Authors response: As pointed by the reviewer, we modified the paper titles and all the journal names, which are not consistently before.

We look forward to hearing from you in due course regarding our submission and to responding to any further questions and comments you may have.

Sincerely,
Ziying Ran